# Carbon Effects from Intra-Product International Specialization: Evidence from China's Manufacturing Industries

Ye Tian [1,2], Wenyu Guo [1,3], Hao Sun [1,4] and Yao Tan [5,6,*]

1   School of Economics and Management, Hubei University of Technology, Wuhan 430068, China; tianye@hbut.edu.cn (Y.T.); guowenyu@hbut.edu.cn (W.G.); sunhao@hbut.edu.cn (H.S.)
2   Hubei Circular Economy Development Research Center, Wuhan 430068, China
3   Hubei International Trade Barrier Response Research Center, Wuhan 430068, China
4   Hubei Innovation Research Center of Rural Social Management, Wuhan 430068, China
5   School of Finance, Nankai University, Tianjin 300381, China
6   CRCC Financial Leasing Company, Tianjin 300457, China
*   Correspondence: tayao@netease.com

**Abstract:** Intra-product international specialization promotes the global diversification of manufacturing industries with various carbon intensities. With the emerging topic of global warming, a new constraint on the use of carbon in international trade is being imposed on developing countries such as China. To explore the potential effects of this constraint on the progress of specialization, a new theoretical framework was proposed with a series of empirical tests derived from detailed panel data built on statistics from 2004 to 2020 from manufacturing industries in China. The test results indicate that carbon emissions and emission levels in manufacturing industries are partially induced by specialization. Industries with various attributes present heterogeneous performances under the carbon effect. Intra-product international specialization has more significant carbon effects on certain industries, such as those with a limited technique, capital-intensive industries, and industries that use a medium to a high level of carbon. Therefore, given the carbon constraints, high-quality development in manufacturing industries may be attained in developing countries such as China through improvements in specialization in the international market and incremental foreign investment in high-value-added and low-carbon production sectors. These improvements could be secured by implementing appropriate industrial policies and constraints on energy consumption.

**Keywords:** carbon effects; intra-product international specialization; share of vertical specialization (VSS)

## 1. Introduction

Intra-product international specialization is a global process of spatial redistribution with various production units in a given product, which is the next step in international specialization after inter-industry and intra-industry specialization, resulting in the optimization of resource allocation and promotion of international trade. With the growth in international trade, international specialization has established firm ties to ecological and environmental factors, as a series of environmental issues were found, e.g., the cost of resource supplies and pollution emissions increased, environmental quality deteriorated, and global warming accelerated. Following international specialization and relevant global investments, these issues are attracting more attention worldwide. In recent years, developing countries (e.g., China) have begun to notice the potential effects of global warming due to international specialization and have consequently transformed their domestic industry. The 20th CPC national congress report emphasized that "Reaching peak carbon emissions and achieving carbon neutrality will mean a broad and profound systemic socio-economic transformation". Based on China's energy and resource endowment, we will advance initiatives to reach peak carbon emissions in a well-planned and phased way, in line with the

principle of building the new before discarding the old. We will exercise better control over the amount and intensity of energy consumption, particularly of fossil fuels, and transition gradually toward controlling both the amount and intensity of carbon emissions. (The report of 20th CPC national congress, translated by Xinhua News Agency, China). With Chinese industries' consistent engagement with international specialization, the publicized target of "Carbon Peaking and Carbon Neutrality" is more than an environmental policy: it indicates the potentially important interests of all developing countries that aim to achieve an advantageous position in global competitions under the current specialization models, including India, Vietnam, and Mexico. To understand why these interests are so important for developing countries, it is important to determine how the development and transformation of industries in developing countries influence carbon emissions, as well as whether there is heterogeneity in carbon emissions in different industries affected by intra-product international specialization. Since China is one of the largest developing countries, with many diversified industries that contribute to the identification of heterogeneity, this article selected China as the research object.

There are two primary pathways by which intra-product international specialization may influence carbon emissions caused by production. The first pathway is the model of specialized production. As an extended process of further specialization in production, the comparative advantages of intra-product international specialization would increase investment and trade by stimulating the intra-product trade globally, which would consequently change the spatial distribution and relevant quantity of carbon emissions. The second pathway is the clustering and transfer of the industries producing an intermediate level of goods, signifying that intra-product international specialization differs from inter-industry and intra-industry specializations. The pathway specifies how carbon emissions change during specialization (as shown in Figure 1). This article focused on the second mechanism of influence, the progress of specialization spatially redistributing the various procedures and production units to different regions. Procedures and production units with intensive carbon would eventually be removed from the final goods. The removal progress may invoke the clustering and transfer of intermediate goods in various countries according to their endowment and regulations, which implies diverse levels of carbon emissions. If the carbon-intensive units are clustered in a country, the carbon emissions of this country would probably increase. In contrast, a country may significantly reduce its carbon emissions by transferring most high-carbon production units overseas. There is another possibility of a two-way diffusion in which parts of the carbon-intensive units are clustered domestically while others are transferred overseas, which achieves ambiguous results.

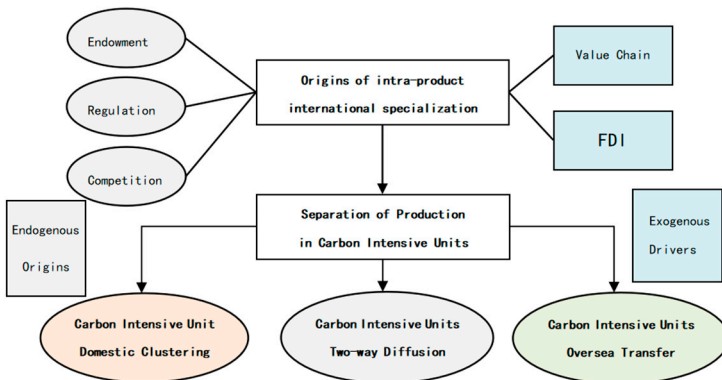

**Figure 1.** Mechanism of intra-product specialization on carbon emissions from production [1].

In the past 30 years, academic research has been devoted to revealing the nature of economic growth, international specialization, and the consequent ecological environment. Early studies analyzed the effects of manufacturing on industrial pollution. With increasing attention to global warming and international specialization, carbon emissions from manufacturing have become one of the core issues in this area. Table 1 displays the primary

research areas and the notable literature on previous studies that are pertinent to the topic. Previous studies may be improved in the following three parts: (1) The applied theoretical frameworks lag behind the advancement of contemporary thinking. Research lacks support from systematic and various theoretical models, most of which employ the traditional theories of inter-industry and intra-industry specialization as frameworks, missing the analysis of the intra-product perspective. (2) Most analyses are given from the perspective of the traditional three effects, i.e., scale, composition, and technique effects, which bind with international specialization to create a new relation that few have noticed. (3) Few articles reveal the mechanism by which international specialization influences the emissions of pollutants or carbon as issues at the micro-level are rarely investigated.

**Table 1.** Related literature by topic.

| Topic | Core Viewpoints |
|---|---|
| The externality of international specialization | |
| damage from specialization | These arguments emphasized the spreading of pollution caused by international specialization and concurred with the pollution heaven hypothesis as well as the "race to the bottom" theory [2,3]. |
| benefits of specialization | These arguments emphasized the diffusion of creative and environmentally friendly technologies [4,5]. |
| complex process | This perspective highlighted the potential presence of an Environmental Kuznets Curve (EKC) and suggested that the impact of international specialization on ecological degradation varies according to spatial and temporal factors [6–10]. |
| Effects of international specialization on the environment and its quantification | |
| ACT model | This model offers a practical approach to examining the three environmental effects and evaluating the influences of international trade on the environment. Many studies have adopted this model or its improved counterpart [11–15]. |
| Input–output model (IO) | Input–output models, which are derived from the theory of equilibrium and reproduction, are able to examine the connection between sectors in terms of supply and demand. Their theoretic fundaments provide broad applications beyond economic systems [16–19]. |
| Simultaneous equations model (SEM) | The simultaneous equations model is created by incorporating the principles of trade and growth, as well as the environmental Kuznets curve, to expand the analysis of the three environmental effects [20–22]. |
| Action mechanism of international specialization on environmental pollution | |
| International trading perspective | Previous research discussed three primary viewpoints. The first perspective involved examining how specialization affected the environment by comparing trade patterns across various countries. The second perspective aimed to understand the role of trade policy in the three effects and explores their mechanisms. The third perspective delved deeper into analyzing the mechanisms behind the individual effects [23–28]. |
| Industrial perspective | Previous research has been categorized into three primary viewpoints: first, examining the distinct operational mechanisms of various industries from an industry standpoint; second, investigating the impact of foreign investment on environmental pollution from an asset perspective; and third, exploring the connection between the international specialization within companies and environmental pollution from a micro-level standpoint [29–33]. |

This article concentrated on the effects of a particular form of specialization, i.e., intra-product international specialization, on the carbon emission from production by constructing a theoretical model by empirically analyzing relevant data from China. Policy suggestions were provided on how developing countries may balance their industrial development with low carbon emissions under consistent engagement in intra-product international specialization. The research design for this article is as follows (Figure 2):

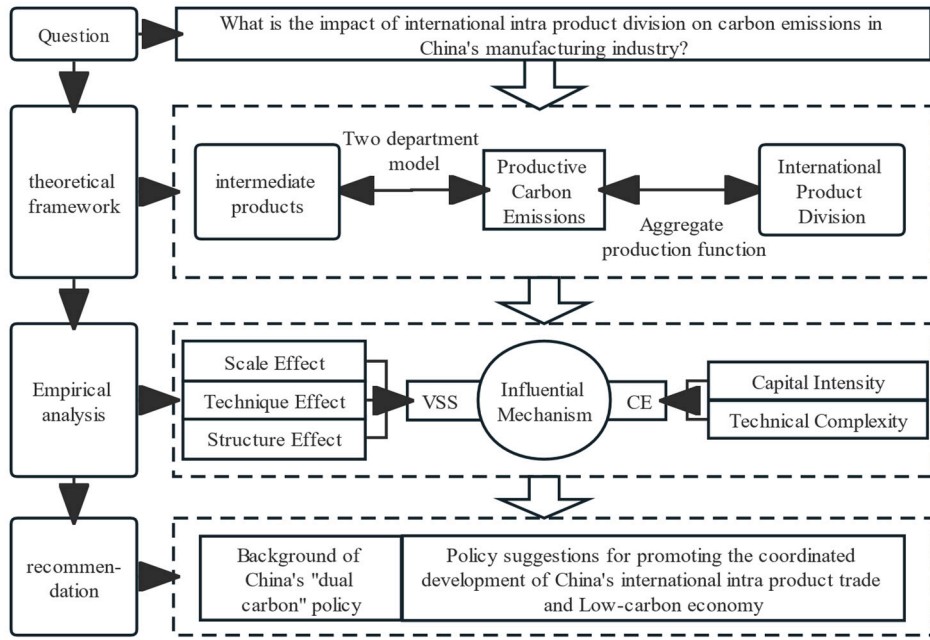

**Figure 2.** Research design.

There are 6 sections in this article. Section 1 is the introduction. Section 2 is the literature review. Section 3 presents the theoretical framework of the empirical analysis. Section 4 presents the estimation of carbon emissions in each manufacturing industry, as well as the level of intra-product international specialization. In this section, a quantitative model is constructed to analyze the relationship between carbon emissions and intra-product international specialization. Applying this model, the empirical study evaluates the effects of intra-product international specialization on carbon emissions using the 2004–2020 panel data of China's specific manufacturing industries and other statistical results. Other relevant issues of estimations are presented, including the model of estimation, variable selections, data sources, and necessary treatments. Section 5 presents the series of estimated results with comments. Section 6 includes the conclusion and policy suggestions.

Compared to the previous literature, the innovation of this article is the incorporation of the intra-product trade model into the study of international trade and environmental issues. Most of the previous literature used the conventional theory of inter-industry and intra-industry international specialization as the analytical framework, with less emphasis on the perspective of intra-product specialization. The article signifies the following aspects: (1) Abandoning traditional inter-industry specialization theories based on comparative advantages, a new theoretical framework is constructed on intra-product international specialization. (2) The article evaluated the level of intra-product international specialization by applying the share of vertical specialization (hereafter referred to as VSS) and input–output analysis. With statistical data, the empirical study in this article approved the effects of intra-product international specialization on domestic carbon emissions from production, with a further exploration of heterogeneity.

## 2. Literature Review

International specialization is fundamental for a nation's participation in global trade and the world economy. The study of specialization in economics has evolved from the traditional theory of comparative advantage to more subtle specialization within industries and even within specific products. The former generally stresses the specialization among industries. The concept of specialization refers to the geographical allocation of various processes, segments, and components involved in the production of a specific product in different countries. All these countries are parts of the value chain for a given product and specialize in producing specific segments [34]. This type of specialization is influ-

enced by factors such as technology [35], barriers [36], economies of scale [37], and firm boundaries [38], and promotes production and trade. Meanwhile, it delivers environmental impacts as externalities. The primary interest points of current academic research on international specialization and the ecological environment are grouped into the following three areas.

### 2.1. The Externality of International Specialization

There are three typical arguments about the externalities of international specialization in the ecological environment. The first argument claims damage from specialization. The argument emphasized the spreading pollution brought by international specialization and concurred with the pollution heaven hypothesis as well as the race to the bottom theory [2,3]. The second argument highlighted the benefits of specialization, generally caused by the diffusion of creative and environmentally friendly technologies [4,5]. The last argument regards this issue as a complex process. Previous empirical studies proved the effect of specialization on the environment may present an environmental Kuznets curve (EKC), where the deterioration of the environment is driven by international specialization in a revert "U" shape [6]; other studies found that the relationship between the above two may be concluded in other forms, including "U" shape [7], "N" shape [8], and cubic equation [9].

### 2.2. Effects of International Specialization on Environment and Its Quantification

Grossman and Krueger (1993) [11] proposed that the total environmental impact could be broken down into three specific parts, i.e., scale effect, composition effect, and technique effect, which were applied to empirical studies with several models developed by the following researchers. (1) The first one is the Antweiler–Copeland–Taylor model (ACT), named after its developers [12]. The model has been improved by several researchers [39,40]. Most applications of the ACT model proved significant effects on carbon emissions from international specialization and relevant trading production. But some of the applications observed potential reductions in emissions [14,15] and others indicated the existence of a pollution heaven [13,41]. (2) The second model is the input–output model (IO). Due to the limited accessible data, areas in empirical studies may not be expanded to territories beyond the investigators' own country or region. A research model with only one region is called single region input–output (SRIO) [16,17]. Models for multiple regions are called input–output models (MRIO) if there are sufficient data [18,19]. (3) The third is the simultaneous equations model (SEM), which is widely used for the total environmental effects and its divided parts in empirical studies with international specialization. Most studies found a positive influence on environmental quality for the scale effect and a negative influence on the technique effect, while common acknowledgment for the structure effect is absent [21,22]).

### 2.3. Action Mechanism of International Specialization on Environment Pollution
#### 2.3.1. International Trading Perspective

Previous research has concentrated on the following three categories: (1) The first is the forms of specialization and trading patterns. Most studies analyzed inter-industry trading patterns based on the theory of comparative advantage [23,42]. Others have investigated potential effective pathways and environmental consequences in various forms of specialization, including the processing trading pattern derived from vertical and intra-product specializations [24,43], global value chain [25], and international outsourcing [26]. (2) The second is trade policy. This category involves various studies, including environmental policies related to tariff barriers and export subsidies [44], environmental effects of regional economic cooperation [27], and environmental outcomes of other policies and regulations [45]. (3) The third is the specific consequences of the broken-down effects. Studies on technique effects focus on exploring the pathways of technique spillover and transfers caused by international trade [46], and the relationship between carbon reduction

efficiency and international trade [47]. Composition-effect-related studies tried to figure out the mechanism of action and quantify effective levels [48].

### 2.3.2. Industrial Perspective

Previous studies have addressed the following issues: (1) The first is the environmental effects in a specific industry. Most articles on this topic tried to establish relationships between manufacturing and trade in pollution- and carbon-intensive industries to those environmental consequences, as well as to potential governmental regulations [49]. It is argued that environmental regulations may raise the cost and lower productivity and competency [29]. In contrast, others indicated enhancements in creativity and competency from environmental regulations [50]. Previous studies also displayed a non-linear relationship between the regulations and export competency, which was influenced by regional levels of economic development [30], market competition [51], and the level of urbanization [52]. (2) The second is the functions of FDI including the participation of foreign capital [53], the relationship between FDI and the transfer of polluting industries [54], location selection and space distribution of industrial transfer driven by FDI [55], and environmental effects from technique spillovers and transfers with FDI [31]. (3) Corporate investigation. Few studies looked into corporate operations. Representative studies include the analysis of the corporate nature [56], and the application of heterogeneous-firm trade theory to firms under general specialization [57].

## 3. Theoretical Framework

### 3.1. Effect of Intermediate Good on the Carbon Emission of Final Good

Assuming there is one final good Y produced in a given developing country, which requires z types of intermediate goods, i.e., type $x_1 \sim x_z$, the only intermediate good produced domestically after intra-product international specialization is the one with the sole input of labor under a monopolistic competition market, while other intermediate goods rely on imports from developed countries. Intermediate goods can be classified into two categories according to their carbon emissions during production: the high carbon category symbolized by $\alpha$, and the low carbon category symbolized by $\beta$. The high carbon category may have negative effects on global society. The corresponding final goods produced from these intermediates can be described by the production function as follows:

$$Y = X_\alpha^\sigma X_\beta^{1-\sigma} \tag{1}$$

where $Y$ is the quantity of final good, and $X_1$ and $X_2$ are, respectively, the summarized quantities for the two categories of intermediate goods, $x_{1j}$ and $x_{2j}$, with a symmetric form in the equation. According to Either (1982) [58], the production function of intermediate goods can be described by the constant elasticity of substitution (CES) as follows:

$$X_\alpha = \left[\sum_1^m x_{\alpha j}^{(e-1)/1}\right]^{e/(e-1)} \text{and } X_\beta = \left[\sum_1^n x_{\beta j}^{(e-1)/1}\right]^{e/(e-1)}, e > 1 \tag{2}$$

where e is the price elasticity of demand for intermediate goods, which is a constant. Equation (2) indicates that the return of the final goods grow with the scale of the number of intermediate types in either the high carbon category m or the low carbon category n. Given that the only input of intermediate goods in developing countries is labor with free mobility, the cost function of the two categories of intermediate goods may be written as follows:

$$C_{ij} = a + bx_{ij}, \ i = \alpha, \beta \tag{3}$$

where $C_{ij}$ is the quantity of labor input for the production of type j intermediate goods in category *i* and *a* is the fixed cost as the constant marginal cost. Assuming that the cost functions of all types of intermediate goods are identical, and monopolistic competition increases the marginal costs of enterprises, we obtain $wb = P_i(1 - 1/e)$, where w is the

ratio of wages and $P_i$ is the price of intermediate goods. Then, we obtain the price function at maximum profit as follows:

$$P_{ij} = wb(e/e - 1) \tag{4}$$

In the standard Dixit and Stiglitz model of monopolistic competition, the free-entering barrier with returns to scale leads to zero profit at equilibrium. Let the profit of the representative enterprise be zero and let Equation (4) be substituted. We obtain $\pi_{ij} = wb(e/e - 1)x_{ij} - w(a + bx_{ij}) = 0$, from which the output quantity of each type of intermediate good can be solved, given that the cost functions and elasticities are identical for all production departments of intermediate goods. The result of the solution is as follows:

$$x_{ij} = (e - 1)(a/b) \tag{5}$$

Equations (4) and (5) provide the price and quantity of intermediate goods.

We assume that the utility function of the representative consumer is

$$U = {(Y/L)^{1-\rho}}/{1 - \rho} - \gamma \cdot CE \, , \, \rho > 1, \gamma > 0 \tag{6}$$

where $\gamma$ is the negative marginal utility of carbon emissions for the society, $\rho$ is the elasticity of marginal utility in consumption per capita, and $CE$ is the quantity of carbon emissions. Assuming that all the income of consumers is paid for the final goods, the budget constraint function is as follows:

$$w \cdot L + \tau \cdot CE = P_Y \cdot Y \tag{7}$$

where $P_Y$ is the price of the final good. The total income consists of two parts: the returns of labor input $w \cdot L$, and tax for carbon emission $\tau \cdot CE$, with the taxation rate $\tau$ measuring the level of carbon regulation. Revenue from the carbon tax is positively proportional to the quantity of carbon emissions, $CE$. The carbon tax is assumed to be returned to consumers after taxation by the government.

Assuming that the carbon emissions in the production of final goods are ignored, the carbon emission level of final goods is determined by the production of intermediate goods, whose carbon emission function is as follows:

$$CE = m^{e/(e-1)} x_1 \theta^{-\mu} \tag{8}$$

where n is the type number of high-carbon intermediates, $\theta^{-\mu}$ is the factor of pollution emission, and $\mu$ is the efficiency of carbon reduction or capture. The parameters in the equation satisfy $\partial CE/\partial \mu < 0$ and $\partial^2 CE/\partial^2 \mu < 0$.

Given a closed economy, the number of intermediate goods in each category of carbon intensity is solved by applying the constraint of maximum profit for the final good. According to Equation (1), the portion of $x_\alpha$ in the total cost is $\sigma$, while the portion of $x_\beta$ is $1 - \sigma$. Given Equation (7) as a budget constraint, we obtain the demand functions of $X_\alpha$ and $X_\beta$ in developing countries as follows:

$$X_\alpha^D = \frac{\sigma P_Y \cdot Y}{P_\alpha + \tau} = \frac{\sigma(w \cdot L + \tau \cdot X_\alpha)}{P_\alpha + \tau} \tag{9}$$

$$X_\beta^D = \frac{(1 - \sigma)P_Y \cdot}{P_\beta} = \frac{(1 - \sigma)(w \cdot L + \tau \cdot X_\alpha)}{P_\beta} \tag{10}$$

The corresponding supply functions of intermediates in these two categories can be inferred by Equations (2) and (5), along with the relevant predefined types *m* and *n*. In this case, the supply functions for developing countries are as follows:

$$X_\alpha^S = m^{e/(e-1)} x_\alpha = m^{e/(e-1)} \frac{a}{b}(e - 1) \tag{11}$$

$$X_\beta^S = n^{e/(e-1)}x_\beta = n^{e/(e-1)}\frac{a}{b}(e-1) \tag{12}$$

The type number of intermediate goods m and n at equilibrium are given as follows:

$$m = \left[\frac{\sigma L P_\alpha}{\sigma e(\tau(1-e) + P_\alpha)}\right]^{(e-1)/e} \tag{13}$$

$$n = \left[\frac{(1-\sigma)L(\tau + P_\beta)}{\sigma e(\tau(1-\sigma) + P_\beta)}\right]^{(e-1)/e} \tag{14}$$

To determine the relationship between carbon regulation and type number in each carbon category, Equations (13) and (14) are differentiated with respect to $\tau$ separately. Here we obtain the following:

$$\frac{\partial \mathrm{m}}{\partial \tau} < 0 \text{ and } \frac{\partial \mathrm{n}}{\partial \tau} > 0 \tag{15}$$

Equation (15) indicates that the marginal increase in carbon tax $\tau$ is going to lead to less usage of high-carbon intermediates in final goods and more usage of low-carbon intermediates. Given the other unvarying conditions, the carbon emissions of final goods are determined by the intermediate goods.

### 3.2. Influence of Intra-Product International Specialization on Carbon Emission

After applying the above theoretical analysis to the improved classic supply–demand model of environmental pollution by Copeland and Taylor (2004 [59]), a general equilibrium model can be established on the condition of intra-product international specialization with carbon emission. We assumed that there is a country with the following presumptions: (1) The country contains only two sectors, i.e., production sectors and research sectors. *Y* is the final good and *X* is the input of intermediate goods driven by the intra-product international specialization, where X is from import. (2) The production factors are restricted to only two factors, i.e., labor and capital. (3) The environmental pollution during production is ignored to obtain a simplified model for the study. (4) The returns to scale are constant.

The carbon emission allowance can be given by analyzing the above model. Since the governmental regulation of carbon emission is intrinsic, the increasing requirement of low carbon society by consumers with the growth of per capita income urges the government to raise the standard of carbon regulation. From the governmental perspective, the selection of regulation standards is a decision given at the optimized carbon emission level, which best meets the demand of citizens' welfare. The optimized carbon emission level is given as follows:

$$\max_{CE}\{V(I/P) - \mu \cdot CE, \qquad s.t.I = G(EN, P, CE)/N\} \tag{16}$$

where the indirect utility function *V* is given by real income and carbon emission (carbon emission allowance supply) (In Copeland and Taylor pollution supply model, the welfare level of representative residents may be depicted by the indirect utility function: *V* = *u*(*I/p*) − *μ·CE*, where *V* is the first-order linear equation of real income and pollution level), *I* is resident income, and *P* is the price of goods. Then, *I*/*P* is the index of real income level, *CE* is the carbon emission level, and *μ* is the utility coefficient for carbon emission. *G* is the given function of factor endowment EN, price of good P, and carbon emission level *CE*. *N* is the number of residents. From $I = G(EN, P, CE)/N$, both sides divided by *P* obtain $I/P = G(EN, P, CE)/NP$, which is substituted to Equation (9). Then, the first-order condition of the composite function is obtained by applying the first derivative to the substituted function:

$$V_{CE}(I/P) = V'(I/P) \cdot G_{CE}(EN, P, CE)/NP = \mu \tag{17}$$

From (10), we obtain the following:

$$G_{CE}(EN, P, CE) = \mu NP/V'(I/P) \tag{18}$$

where $G_{CE}$ is the marginal revenue of carbon emission, and $\mu NP/V'(I/P)$ is the marginal cost of carbon emission. The optimized supply of carbon emission is given at the condition where marginal cost equals marginal revenue, which also indicates the condition where the maximized residents' utility is achieved by controlling the carbon emission.

$$Y = (1-r)G\left(L_e, n, K_d, K_f\right) = (1-t)A[L_e(1-n)]^{\alpha}K_d{}^{\beta}K_f{}^{\gamma} \tag{19}$$

where $A$ is the constant, $L_e$ is labor productivity, $n$ is the proportion of research staff to the total labor, $1-n$ is the proportion of production staff to the total labor, $K_d$ and $K_f$ are capitals from domestic and foreign sources, respectively, and $\alpha + \beta + \gamma = 1$; $r$ is the allowance of carbon emission under regulation, inferring that the $r$ proportion of production is used to invest in carbon emission reduction by the government. Let $r = ke(\theta)$, where $k$ is the fixed coefficient and $e(\theta)$ is the intensity of regulation. Substituting this equation of $r$ to Equation (19), it is obtained that $Y = (1 - ke(\theta))A[L_e(1-n)]^{\alpha}K_d{}^{\beta}K_f{}^{\gamma}$, which indicates the relationship between carbon emission and production, where intensive carbon emission could suppress economic growth. Then, the carbon emission level is given as follows:

$$CE = (1 - ke(\theta))G\left(L_e, n, K_d, K_f\right) = CE(\theta) \cdot G\left(L_e, n, K_d, K_f\right) \tag{20}$$

where $CE(\theta) = 1 - ke(\theta)$, measuring the technique's effect on carbon emission. As the production of the final good $Y$ requires the input of intermediate good $X$, it is assumed that $x$ is the input of intermediate good $X$ driven by intra-product international specialization, $y$ is the input of intermediate good driven by others excluding intra-product international specialization, and $\rho = \rho_{(x,y)}$ is the level of intra-product international specialization. The equation can be rewritten as $CE = (1 - ke(\theta))G\left(L_p, n, K_d, K_f\right) = CE(\theta) \cdot CE(\rho) \cdot CE(s) \cdot CE(p)$, indicating the four type effects, i.e., technique effect, intra-product international specialization effect, scale effect, and composition effect. By applying derivatives of time t to both sides in Equation (19), we obtain:

$$\frac{dCE}{dt} = \frac{dCE(\theta)}{dt} \cdot CE(\rho) \cdot CE(s) \cdot CE(p) + CE(\theta) \cdot \frac{dCE(\rho)}{dt} \cdot CE(s) \cdot CE(p) + CE(\theta) \cdot CE(\rho) \cdot \frac{dCE(s)}{dt} \cdot CE(p) + CE(\theta) \cdot CE(\rho) \cdot CE(s) \cdot \frac{dCE(p)}{dt} \tag{21}$$

where $\frac{dCE}{dt}$, $\frac{dCE(\theta)}{dt}$, $\frac{dCE(\rho)}{dt}$, $\frac{dCE(s)}{dt}$, and $\frac{dCE(p)}{dt}$ are, respectively, changes in carbon emission, technique level, level of intra-product international specialization, scale level, and composition level, which unveil the tight relationship between carbon emission and the changes in the four types of effects. The measurements of these four types of effects on carbon emissions are quantified in the following section.

## 4. Methodology and Data

### 4.1. Research Model

For the purpose of measuring the influence of intra-product international specialization on the carbon emission in China's manufacturing industries, the following quantitative regression model was employed with China's panel data from 2004 to 2020 (15 consecutive years) covering 26 industries.

$$CE_{it} = \alpha_0 + \alpha_1 VSS_{it} + \alpha_2 controls_{it} + \varepsilon_{it} \tag{22}$$

where subscript $i$ and $t$ are industry and year; $CE_{it}$ is the carbon emission level at year $t$; and $VSS_{it}$ is the index of intra-product international specialization in the manufacturing industry and the key explanatory variable, which is given by measuring the trade of

intermediate good in the industry $i$ at year $t$. *controls$_{it}$* are a series of industry-relevant controlled variables.

### 4.2. Variables and Data

### 4.2.1. The Measurement of VSS

The measurement of intra-product international specialization is crucial to supply the quantified results of the research model. There are two major measuring pathways as follows. The first pathway provides the index of specialization with the statistics from international trading, initially proposed by Ng and Yeats (2003) [60], who presented the index by calculating the revealed comparative advantage (RCA) with the data collected in the trade of parts and components. Following this pathway, the predominantly used indexes are the price index for exported goods and the complexity of exported techniques. As the former substitutes the differences in exported goods with the prices of goods, the result obtained explicitly reflects the position in the global value chain. The latter produces the index of specialization with the data covering long periods, which provides a solid foundation for the analysis of the productivity of exported goods, the complexity of exported techniques, and the comparative advantage of goods. The second pathway measures the level of international specialization through the input–output tables, which provide the index of specialization by constructing a matrix with multiple input–out tables of various countries. The typical methods in this pathway are the GVS index and VS index. The VS index may be further divided into two types, i.e., the absolute VS index and the relative VSS index. The absolute VS index measures the value of imported intermediate goods that are supplied to exported goods. The relative VSS index is the proportion of the value measured in the VS index in terms of the total export value.

Since the input–output tables are more precise reflections of the trade of intermediate goods (an inevitable flaw exists in measuring vertical specialization with international trading; the data are easy to access but contain no information about the inter-relations among various production phases, which is not able to explicitly depict the phenomenon of how vertical specialization originates from the production process.), the VSS index is employed to evaluate the level of vertical specialization in China's manufacturing industries, which indicates the participation of the Chinese industries in intra-product international specialization. Compared to the index given by the proportion of processing trade in total trade, the estimation of VSS is more meaningful to illustrate the participation of manufacturing industries in intra-product international specialization. This article used the input–output data from a single country (China) to estimate the index, which may be not enough to fully present the effect of the trade of intermediate goods on final goods. In this case, the application of multiple input–output analysis could provide a more accurate estimation of intra-product specialization. But this article could not use it due to the limited accessible data. The VSS index is calculated as follows:

$$VSS = \frac{1}{X} u A^M (I - A^D)^{-1} X^V \tag{23}$$

where, $u = (1, 1, \cdots 1)$; $A^M = \begin{pmatrix} a_{11} & \cdots & a_{1n} \\ \vdots & \ddots & \vdots \\ a_{n1} & \cdots & a_{nn} \end{pmatrix}$ is the exemplified coefficient matrix of

intermediate goods; $I$ is the unit matrix; $A^D = \begin{pmatrix} b_{11} & \cdots & b_{1n} \\ \vdots & \ddots & \vdots \\ b_{n1} & \cdots & b_{nn} \end{pmatrix}$ is the coefficient ma-

trix of domestic consumption of intermediate goods; $X^V = \begin{pmatrix} X_1 \\ \vdots \\ X_n \end{pmatrix}$ is a vector of export;

$\left(I - A^D\right)^{-1}$ is the Leontief inverse matrix; $n$ is the number of industries in each manufacturing sector.

Given that the Chinese input–output table was updated every five years, data in the gap years were estimated according to the published tables in seeking continuous data. The estimation method used the minimum of errors in the weighted quadratic objective function to infer the direct consumption coefficient of manufacturing industries in 15 consecutive years (2004–2020). (There are two methods available for estimating the missing input–output table in gap years, i.e., minimum errors and RAS. The method of minimum errors constructs a quadratic function of the direct consumption coefficient, giving the minimum by Lagrange's multipliers, which in fact becomes a problem of Lagrange's indeterminate coefficients. Either method works with a small gap in time, whereas the method of minimum errors provides fewer errors. The Chinese Statistical Bureau published the input–output table every year, i.e., in 2002, 2007, 2012, and 2017. In addition, there are four extended tables published in 2010, 2015, 2018, and 2020. Data in the other years require to be estimated.) The obtained coefficients were applied to the VSS calculation, measuring the level of intra-product specialization in Chinese manufacturing industries. The data used in the calculation were from UNCOMTRADE, a database of the United Nations.

### 4.2.2. Measurement of Carbon Emission

Since the majority of carbon emissions by firms were generated from the primary energy consumption, the carbon emission was calculated according to the primary energy consumption listed in the "Chinese Annual Energy Statistics", including the consumption of coal, coke, crude oil, oil, and natural gas, where all types of primary energy were listed as standard coal.

$$DC_i = \sum_{j=1}^{n} AD_{ij} \times NCV_j \times CC_j \times O_{ij} \tag{24}$$

where $DC_i$ is the carbon emission from the consumption of fossil fuel in industry $i$; $n$ is the number of energy types consumed by industry $i$; $AD_{ij}$ is the type of energy and quantity of energy consumed accordingly; $NCV_j$ is the coefficient of energy type $j$ as listed in standard coal. The heat generated by every unit of a given energy type is $CC_j$, which also represents the coefficient of carbon emission in the given energy type, i.e., the carbon emission in net heat generated by a unit of the energy type $j$; $O_{ij}$ is the efficiency of oxygenation, i.e., the ratio of the oxidized mass during the burning process. Since most coal and oil are used as raw chemicals in the industries with petroleum, coke, and fuel processing of nuclear, the consumptions in these industries are excluded to calculate carbon emissions.

Both the IPCC and the National Development and Reform Commission of China (NDRC) have published default factors ($NCV_j$, $CC_j$) for China. Most of the previous research used the IPCC default values, which were approximately 40% higher than the ones provided in China's on-situ survey [61]. This survey also provided the oxygenation efficiencies $O_{ij}$ for various departments, which would better demonstrate the specific energy utilization efficiency in a given department than a general factor in default. In this case, this article preferentially took the coefficient of energy ($NCV_j$, $CC_j$) along with the oxygenation efficiency $O_{ij}$ in the survey rather than the default ones. Parts of the selected relevant data are shown in Table 2.

**Table 2.** Carbon emission coefficients and standard coal coefficients for the energy types partially selected.

| Coefficients | Raw Coal | Coke | Crude Oil | Gasoline | Kerosene | Diesel Oil | Fuel Oil | Natural Gas |
|:---:|:---:|:---:|:---:|:---:|:---:|:---:|:---:|:---:|
| NCV | 0.21 | 0.28 | 0.43 | 0.44 | 0.44 | 0.43 | 0.43 | 3.89 |
| CC | 26.32 | 31.38 | 20.08 | 18.9 | 19.6 | 20.2 | 21.1 | 15.32 |

Unit: PJ/$10^4$ tonnes, $10^8$ m$^3$, tonne C/TJ.

### 4.2.3. Other Controlled Variables

According to Grossman and Krueger (1995) [62], there are three mechanisms working on socio-economic and pollution, i.e., the total scale of the economy (scale effect), the composition of sectors in society and economy (composition effect), and the quantity of pollution emitted by every unit output (technique effect). In accordance with the above findings, the following controlled variables were considered: (1) gross industrial output value PV, the multiplier of annual rate of industrial added value, producer price index, and the gross industrial output value in the previous year; (2) number of enterprises, NoE, was the sum of the enterprises above designated size and state-holding; (3) number of employees, EMP, was the average of annual employees; (4) productivity of labor, LP, was the ratio of gross industrial output value to the number of average annual employees; (5) fixed asset value, NFA; (6) foreign direct investment, FDI, was the ratio of foreign investment to the asset value of the given industry. All the variables were deducted by inflation and converted to logarithms.

### 4.3. Sources and Descriptive Statistics of Data

This article evaluated the effects of intra-product specialization on carbon emission by constructing a panel with data from 26 manufacturing industries in China from 2004 to 2020. The classification of industries was in accordance with the Chinese national classification of industries (GB/T4754-2017 [63]), published in 2017. As the classification in statistics changed over time before the announcement of national standards, data from the previous statistics required manual processing to match the national standards. In the given case, industries with incomplete data were excluded in this article. After the exclusion, there were 26 selected manufacturing industries. The raw data relevant to these industries were from the Chinese annual industrial statistics, the Chinese annual statistics, the Chinese annual labor statistics, and the statistics base of the Chinese statistical bureau. For the purpose of providing a panel with consecutive and balanced data, linear interpolation was used to generate new values substituted for the missing ones. The descriptive statistics are given in Table 3.

**Table 3.** Descriptive statistics of variables.

| Variable | Symbol | Obs | Mean | Std. Dev. | Min | Max |
|---|---|---|---|---|---|---|
| Carbon emission | lnCE | 442 | 2.821 | 1.672 | 0.074 | 7.538 |
| Level of intra-product specialization | VSS | 442 | 0.182 | 0.049 | 0.072 | 0.342 |
| Labor productivity | lnLP | 442 | 5.74 | 0.62 | 4.112 | 7.538 |
| Gross industrial output value | lnPV | 442 | 4.717 | 1.183 | 0.596 | 7.015 |
| Number of enterprises | lnNoE | 442 | 8.805 | 1.198 | 4.663 | 11.015 |
| Net value of fixed assets | lnNFA | 442 | 3.441 | 1.036 | 0.066 | 5.648 |
| Number of employees | lnEMP | 442 | 5.115 | 1.006 | 0.859 | 7.233 |
| FDI | lnFDI | 442 | 0.523 | 0.457 | 0 | 2.539 |

## 5. Test Results of the Effect of Intra-Product International Specialization on Carbon Emission

### 5.1. Results of Basic Regressions

As suggested by the Hausman test, the basic regressions were executed with a fixed effect model, which partially solved the endogenous problems resulting from missing variables. Table 4 summarizes the regression estimations of the effects of VSS on carbon emission, CE. The results showed that the coefficients of VSS in all regressions were significantly positive, with most probability values less than 0.05. The addition of extra control variables hardly changed the significant levels of VSS in the regression, indicating the robustness of the model constructed. As the benchmark, column (2) in Table 4 presents the result of regression with a controlled time effect and clustered standard errors at the industrial level, which excluded the unobserved shock in individual industries. The results in column (2) also suggest a positive coefficient of VSS on CE, indicating a 6.8% increase in

carbon emissions caused by a 1% improvement in intra-product international specialization. To further test the robustness of the results, the samples used for the regression were extended with data from 1997 to 2020. The results of the extended samples are shown in column (3), where the coefficient of VSS drops a little with a positive sign. Columns (4) and (5) report the results of the regressions with no control over time. Similar results were obtained for all positive coefficients of VSS, supporting the hypothesis that VSS caused carbon emission. The validated hypothesis was the so-called carbon effect, part of which was triggered by the international intra-product specialization. International intra-product specialization in manufacturing may increase the regional mobility of intermediate goods and their production units. This redistribution process may finally change the effect of productive carbon emissions in each manufacturing industry as a whole. Generally, the level of international intra-product specialization was accelerating carbon emissions in manufacturing industries. In column (2), the coefficient of FDI is negative, indicating a negative relationship between FDI and carbon emissions in manufacturing industries. It was inferred that FDI brought more techniques and composition effects over the scale effect, which caused a negative correlation between FDI and carbon emissions.

**Table 4.** Regression results: carbon emission and intra-product specialization.

| Variables | (1) lnCE | (2) lnCE | (3) lnCE | (4) lnCE | (5) lnCE |
|---|---|---|---|---|---|
| VSS | 5.5144 ** | 6.8288 *** | 4.9729 ** | 2.6735 * | 2.9058 ** |
| | (2.30) | (3.62) | (2.77) | (1.76) | (2.44) |
| lnLP | | −1.5939 ** | −1.3935 *** | −1.1048 *** | −1.1905 *** |
| | | (−2.60) | (−2.99) | (−5.07) | (−5.51) |
| lnPV | | 0.4034 | 0.6519 ** | 0.5973 ** | 0.9160 *** |
| | | (1.48) | (2.35) | (2.45) | (4.47) |
| lnNoE | | −0.3883 | −0.1835 | −0.3004 | −0.0671 |
| | | (−1.29) | (−0.80) | (−1.20) | (−0.62) |
| lnNFA | | 0.6767 ** | 0.6757 *** | 0.4114 * | 0.3249 |
| | | (2.72) | (2.97) | (2.03) | (1.55) |
| lnEMP | | −0.0109 | −0.5656 ** | 0.0005 | −0.5280 ** |
| | | (−0.04) | (−2.41) | (0.00) | (−2.39) |
| lnFDI | | −0.1290 | −0.0853 | −0.0219 | 0.0604 |
| | | (−0.72) | (−0.43) | (−0.12) | (0.31) |
| Constant | 1.6732 *** | 9.7047 ** | 8.8897 *** | 7.1791*** | 6.9472 *** |
| | (3.37) | (2.45) | (2.99) | (3.26) | (5.49) |
| Industry Fixed | Yes | Yes | Yes | Yes | Yes |
| Year Fixed | Yes | Yes | Yes | No | No |
| Obs | 442 | 442 | 618 | 442 | 618 |

Note: *, **, and *** indicate significance levels of 10%, 5%, and 1%, respectively, with robust clustered standard errors in parentheses.

### 5.2. Robustness Test

Although the controls of individual effect and time effect in regression may partially mitigate the missing variable issues, there were inevitable influences from missing variables and sample errors, suggesting more robustness tests required as follows.

### 5.2.1. Substitution of Response Variables

To test the robustness of carbon emission CE, two new variables were used as substitutions, i.e., intensity of carbon emission and quantity of carbon emission. The intensity of carbon emission was constructed by estimating the quantity of carbon emission by every unit of gross industrial output value with primary component analysis. The result of the regression with the newly generated variable is given in column (1) of Table 5, which displays a similar result compared to the previous ones, where intra-product specialization led to the increase in carbon emissions with a positive coefficient of VSS remaining.

**Table 5.** Robustness test.

| Variables | (1) PCA | (2) New Data | (3) Future VSS |
|---|---|---|---|
| VSS | 2.0074 ** (2.11) | 3.6273 * (1.71) | |
| F1.VSS | | | 1.7891 (1.00) |
| lnLP | −0.9234 *** (−3.97) | −1.9409 *** (−3.37) | −1.7322 *** (−2.97) |
| lnPV | 0.0167 (0.10) | 0.7308 (1.54) | 0.7115 (1.62) |
| lnNoE | −0.1314 (−0.76) | −0.0683 (−0.12) | −0.0051 (−0.01) |
| lnNFA | 0.6042 *** (3.18) | 0.8469 * (1.99) | 0.6290 (1.62) |
| lnEMP | −0.2902 ** (−2.41) | −0.0971 (−0.35) | −0.0603 (−0.23) |
| lnFDI | −0.0386 (−0.34) | −0.3040 (−1.16) | −0.2023 (−0.81) |
| Constant | 5.2680 *** (3.39) | 12.5770 ** (2.53) | 11.7725 ** (2.43) |
| Industry Fixed | Yes | Yes | Yes |
| Time Fixed | Yes | Yes | Yes |
| Obs | 442 | 442 | 416 |

Note: *, **, and *** indicate significance levels of 10%, 5%, and 1%, respectively, with robust clustered standard errors in parentheses.

To avoid the inference from the errors in statistics, another new variable of carbon emission CE was produced with the carbon-neutral data in the database of CSMAR from 2004 to 2020. The result of the new variable of carbon emission is given in column (2) of Table 5, which illustrates the VSS coefficient as positive with changing response variables.

### 5.2.2. Placebo Test

The placebo test was executed by substituting the explanatory variable VSS with the variable in one period forward (F1.VSS). Assuming that there was an unobserved inference existing in the regression model, the future VSS may create a positive influence on the current carbon emission. The result given in column (3) of Table 5 presents a positive sign to the VSS coefficient with a probability value over 0.1, indicating the result may not support the assumption at an acceptable confidence interval. In other words, the assumption was probably false, which strengthened the validity of the regressions with limited inference from missing variables.

### 5.3. Endogeneity Test

Various measures were taken to eliminate several possible inferences. The variables related to VSS were generally regarded as exogenous, yet endogenous issues remain in the research. There were two potential endogenous issues. The first one was reverse causality. Given the regulations on carbon emissions, the enterprises sensitive to these regulations would possibly reduce their carbon emission by relocating their manufacturing units to places where carbon emission cost less. The relocating movements of enterprises may result in intra-product international specialization. The second endogenous issue was that the inferences given in the regressions would be the results of missing variables from both carbon emission and intra-product, which weakened the consistency of the estimations. For the purpose of solving these potential problems, two additional tests were introduced to the research model. (1) Substituting the explanatory variables in the benchmark with the variables of lagged one period, i.e., the lagged one-period observations of VSS. The reverse causality may be concluded with the appearance of non-significant coefficients in

lagged VSS, inferring the lagged VSS was indeed influenced by the current carbon emission. According to the previous results, VSS produced significant shocks to carbon emissions, which suggested that the non-significant coefficients of lagged VSS may not be found unless reverse causality exists. The test results are given in column (1) of Table 6 with a significant positive coefficient of lagged VSS, indicating slight reverse causality and the continuous effect of VSS on carbon emissions. (2) Instrumental variables were applied to relieve the inferences from missing variables in the research model constructed. A new instrumental variable was constructed by averaging the VSS of three industries with labor productivities similar to their global competitors. Since the domestic intra-product specialization in a given year was partially connected to its international specialization, the constructed new variable obviously correlated to the original one as required by the instrumental variable. In addition, for any industry, the new variable was generally exogenous as the specialization of intra-product in other industries usually presented little influence on its intra-product international specialization, which was in accordance with the assumption of the exogeneity of an instrumental variable. The result of an instrumental variable was estimated using the two stages least squares (2SLS) method, given in column (2) of Table 6. The first stage coefficient of the instrumental variable was significant and positive, indicating its high correlation to the VSS, which was also proven by the Kleibergen–Paap test, which is a test to identify weak instrument variables, with a Wald F value of 211.89, far over the critical value of 16.38, as suggested by Stock-Yogo. The coefficient of VSS in the second stage regression was still significant and positive, reinforcing the previous conclusion that VSS produces more carbon.

**Table 6.** Endogeneity test.

| Variables | (1) | (2) | (3) |
| --- | --- | --- | --- |
| | Lagged Variable Test | 2SLS Test | LTZ Test |
| VSS | 6.9625 *** | 6.6865 *** | 4.925 ** |
| | (3.71) | (3.93) | (1.99) |
| Control Variables | Yes | Yes | Yes |
| Industry Fixed | Yes | Yes | Yes |
| Time fixed | Yes | Yes | Yes |
| Obs | 442 | 442 | 442 |

Note: **, and *** indicate significance levels of 5% and 1%, respectively, with robust clustered standard errors in parentheses.

Although the above tests provided the validity of the instrumental variable by satisfying the constraints of uniqueness, the exogeneity of an instrumental variable may not be directly observed. Since the equal numbers of both the instrumental variable and the potential endogenous variable, i.e., the original VSS in the benchmark, the over-identification test may not identify the exogeneity. In response to this issue, the robustness inference of LTZ (local to zero approach) given by Conley et al. (2012) [64] was introduced. This approach tested the validity of the coefficient by losing the constraints of uniqueness in a plausible exogenous framework. The result is presented in column (3) of Table 6, with a coefficient value of 4.925 at a 0.05 significance level, which falls into the value range of [0.544, 8.170] at 0.05 confidence intervals, given by the UCI (Union of confidence intervals), indicating the valid exogeneity of the instrumental variable. The above tests further supported the validity of the conclusion.

### 5.4. Heterogeneity Test

Previous studies focused on verifying the correlation between the level of intra-product specialization and carbon emissions. In this article, we further looked into the data related to China's manufacturing industry and tried to identify how the specialization level of a given type of good was related to its carbon emission during production. In this case, heterogeneity tests were required across various types of goods and their international intra-product specializations.

### 5.4.1. Heterogeneity in Technique Complexity

This study separated China's manufacturing industries into low-tech (low_Tech), intermediate-tech (intermediate_Serve), and high-tech (high_Serve) industries based on the industrial R&D input–output ratio, according to the industrial division criterion recommended by the OECD in 1986. The results of the groups' regressions are given in column (1) of Table 7, where the coefficients of VSS present an inverse "V" shape with technique advance. One possible explanation is that the capabilities of absorbing techniques and attributes of various industries result in different productive improvements benefiting from the intra-product international specialization. The participation of technique-intensive industries in international specialization could further improve their productivity with less carbon emissions. Industries with intensive high techniques would have a more fierce international competition, which promotes enterprises in these industries to elevate their productivity and upgrade their techniques. The upgradation of these enterprises could be achieved by pure technique improvement and industrial transformation, which further reduce carbon emissions. Compared to enterprises in high-technique industries, those in low or intermediate-technique industries, especially those in processing trade, are weakly positioned in the global value chain. The participation of those enterprises in international specialization may not contribute to their productivity and carbon emission reductions due to their limited capabilities of research and development. With great sensitivity to the cost of manufacturing and trade, enterprises in low-technique industries tend to invest more in manufacturing to enhance their efficiency of production. Their ignorance of carbon emission reduction results in the unsatisfying effects of carbon emissions.

**Table 7.** Heterogeneity test.

| Variable | (1) Technique Complexity | (2) Capital Intensity | (3) Carbon Intensity |
|---|---|---|---|
| low_Tech | 7.0663 *** (4.05) | | |
| intermediate_Serve | 10.4030 *** (2.73) | | |
| high_serve | −4.6934 * (−1.90) | | |
| low_Capital | | −3.4988 ** (−2.11) | |
| intermediate_Capital | | 7.7617 *** (3.98) | |
| high_Capital | | 10.7332 *** (3.38) | |
| low_Carbon | | | −6.0945 * (−1.85) |
| intermediate_Carbon | | | 5.5610 *** (2.69) |
| high_Carbon | | | 8.2675 *** (2.65) |
| Cross-terms | 2.708 (1.58) | 2.708 ** (2.37) | −1.742 * (−1.67) |
| Control Variables | Yes | Yes | Yes |
| Industry Fixed | Yes | Yes | Yes |
| Time Fixed | Yes | Yes | Yes |
| Obs | 442 | 442 | 442 |

Note: *, **, and *** indicate significance levels of 10%, 5%, and 1%, respectively, with robust clustered standard errors in parentheses. All dependent variables are previous instrumental variables that passed the endogeneity tests.

### 5.4.2. Heterogeneity in Capital Intensity

Similar to the classification in technique complexity, all industries are categorized into three groups according to the net balance of fixed assets in each industry. Here, the net

balance of fixed assets symbolizes capital intensity. The results are shown in column (2) of Table 7, indicating the capital intensity with a positive adjustment effect on carbon emission under the intra-product international specialization. China's relative proportion of capital factors has been increasing and their resource endowments have been fundamentally reconstructed after joining the global trade. The structure of China's international intra-product specialization has changed dynamically. Different from the labor intensity, the development of China promoted industries with bigger capital intensity participating more in international specialization. As the flow of capital is sensitive to international specialization, internationalized industries in intra-product specialization attract more capital than others. The industries with more capital flowing in are inclined to expand their scales, indicating the predominance of scale effect, while the industries of high labor intensity gain more composition effect or technique effect. In labor-intensive industries, the holistic development of the industry is flexible with a short operating cycle, resulting in weak risk resistance. On the one hand, some of the labor-intensive production sectors have begun to shift out of China, contributing to a reduction in the overall scale effect of carbon emissions. On the other hand, some of China's domestic labor-intensive production sectors were forced to undergo technological transformation and reconstruction in order to cope with their diminishing advantages in the international intra-product specialization, thus contributing to the structural or technological effect to a certain extent.

### 5.4.3. Heterogeneity in Carbon Intensity

In this part, all industries were grouped according to their characteristics related to carbon intensity. As portrayed in columns (3) of Table 7, industries with intermediate and high carbon intensity produced more carbon after participation in intra-product international specialization. The possible reasons for these results may be from two processes. First, in an open economy limited by technology, enterprises with higher carbon intensity may not be able to cut the emission of carbon by relieving their reliance on energy and carbon resources. Secondly, industries and their manufacturing units in developed countries with strict regulations like to relocate to developing countries with loose regulations, which causes the industries with high carbon intensity to likely produce more carbon after relocating to regions with bare carbon control. In contrast, industries with low carbon intensity relied little on carbon resources, and it was easier for them to upgrade their composition to further cut carbon emissions using advanced techniques in developed countries. In addition to the above two processes, there was the possibility that some domestic industries with low-carbon appearance would have their high-carbon manufacturing units transferred overseas.

## 6. Conclusions and Suggestions

### 6.1. Conclusions

This article constructed a model of the relationship between carbon emissions from manufacturing and international product division within China's manufacturing sector. The model was based on 2004–2020 Chinese industry-level data and employed a fixed effect model. The empirical analysis of the impact of international product specialization on Chinese manufacturing emissions was conducted based on the results of the empirical analysis. The following conclusions were drawn from the analysis: (1) Part of the carbon emission effects of China's manufacturing production process may be triggered by the international intra-product specialization. The international intra-product specialization in manufacturing may stimulate the regional mobility of intermediate goods and their production units. This redistribution process may finally change the effect of productive carbon emissions in each manufacturing industry as a whole. (2) The level of carbon emission in China's manufacturing industries is tightly associated with the participation of intra-product international specialization, which accelerates the carbon emission in production to some extent. (3) The impact of product specialization within China's manufacturing sector on emissions is different for each industrial type. International product specialization

has a greater impact on emissions in low-technology industries than in high-technology industries, while its effect on emissions in high-technology industries is more significant. The impact of domestic product specialization is more significant in low-carbon industries than in high-carbon industries, but its impact level is weaker in low-carbon industries than in high-carbon industries.

*6.2. Policy Suggestions*

(1) Increasing the level of international intra-product specialization and accelerating the upgrading process of intra-product trade: In the progress of China's transformation of foreign trade, under the premise of fully understanding and fully accessing the current domestic endowment advantages, it is necessary to gradually enhance the structure of endowments to achieve dynamic transformation of comparative advantages. Moreover, it is necessary to improve the technology intensity of intermediate goods, thus extending China's intra-product international specialization to a higher level. Furthermore, with the comprehensive consideration of carbon regulation with trade efficiency, the industrial policies of those manufacturing industries with deep participation in the international intra-product specialization should be reconsidered. China needs to improve its extensive pattern of economic growth, characterized by blind expansion of production scale, high input, high consumption, and high pollution, and adjust the industrial structure of the high-carbon manufacturing industries. In this way, China could achieve high-quality development of processing trade under the background of "Carbon Peaking and Carbon Neutrality".

(2) At present, the manufacturing structure of Chinese processing trade, which is mainly composed of foreign-invested enterprises, has begun to change. China should try to guide foreign investment into production units with low-carbon, high-value-added, and technology-intensive industries. In this way, China would not only join the production chain of high-tech and high-value-added at an early stage in the international specialization to obtain greater trade benefits but also promote the low-carbon composition effect of intra-product international specialization.

(3) China should accelerate technological innovation and enlarge the investment in research and development of technologies in energy saving and emission reduction. Given the increasing carbon tariffs, technological change is the key to realizing China's carbon target, and China should improve the technique levels in those industries that participated in intra-product international specialization, enhance the technique level and added value of their products, and reduce their consumption of raw materials and energy with lower emission in the production of intermediate goods. China may achieve a high quality of development by driving the alteration of their primary production units from high carbon to low carbon within the intra-product international specialization.

(4) China should set up reasonable regulation intensity and improve their institutional construction of carbon regulation. Due to the heterogeneity of the carbon effect in manufacturing industries, the intensity of regulation should be imposed heterogeneously according to the industries' attributes and their participation in intra-product international specialization. The regulation should give enough consideration to the intermediate good and production units before implementation. The market adjustment and incentive may be flexibly introduced to the design of regulations on intermediate goods or production units, which can eventually promote the efficiency of carbon regulations and the progress of the "Carbon Peaking and Carbon Neutrality" policy objectives of the government.

**Author Contributions:** Conceptualization, Y.T. (Ye Tian) and H.S.; methodology, Y.T. (Ye Tian); software, Y.T. (Ye Tian); validation, W.G.; data curation, W.G.; writing—original draft preparation, Y.T. (Ye Tian) and Y.T. (Yao Tan); writing—review and editing, Y.T. (Yao Tan). All authors have read and agreed to the published version of the manuscript.

**Funding:** This research was funded by the national social science fund of China, grant number 19CJY019.

**Data Availability Statement:** China's input–output tables are from the Chinese statistical bureau. The trading data are from UNCOMTRADE. The industry-specific data are from the Chinese annual industrial statistics, the Chinese annual statistics, the Chinese annual labor statistics, and the Chinese statistical bureau. The carbon-neutral data are from CSMAR.

**Conflicts of Interest:** The authors declare no conflict of interest.

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
