# Peer review of "Carbon Effects from Intra-Product International Specialization: Evidence from China’s Manufacturing Industries"

_sustainability, doi:10.3390/su151612433_

Round 1

Reviewer 1 Report

To explore the potential effects of this constraint under the progress of specialization, a new theoretical framework was proposed with a series of empirical tests through detailed panel data built on statistics from 2004-2020 manufacturing industries in China. However, there are some shortcomings that need major revision. I put forward some suggestions as follows.

1. This paper lacks elaboration on innovation, which cannot highlight how this study differs from the existing literature.

2. A figure of the research design is needed.

3. The paper lacks a discussion part, which cannot highlight the marginal contribution of the results of this study to the existing research.

4. More relevant studies especially recent ones should be included in the literature review and discussion part.

5. There are some grammar errors. Please check carefully.

6. The use of English tense and English font format in this paper is rather confusing. The normative nature of language needs to be strengthened.

7. The policy recommendations in the research conclusions are slightly insufficient. Therefore, it is suggested that the authors put forward more valuable policy suggestions according to the research results.

Reviewer 2 Report

I think This paper has scientific potential and an analytical component. This paper constructs a model of the relationship between carbon emissions from manufacturing and intra-product international specialization within China's manufacturing industries. The impact of intra-product international specialization is provided the based on the results of the empirical analysis. And suggestions are given on how developing countries may balance industrial development with low carbon emission under the consistent engagement in intra-product international specialization.

Further comments and suggestions are following:

1. Introduction. The definition/description of intra-product international specialization of paper is not clear and difficult to understand. The introduction introduces two primary pathways that intra-product international specialization may impose its influence on carbon emission from production, but the theoretical framework and the following paper mainly discuss the second mechanism, ignoring first mechanism.

2. Literature review. This part lacks related researches on intra-product international specialization, and the innovation points of the article are not pointed out.

3. Heterogeneity Test. (1) It needs to be pointed out whether endogeneity is considered in the part of Heterogeneity Test. (2) and in the part of “Heterogeneity in technique complexity”, the specific basis for division should be given in detail. (3) Whether the coefficients shown in Table 6 represent the effects of dummy variables representing different groups or the coefficients of VSS in different groups should be clarified in the text to avoid confusion.

4. The results shown in the column (2) of table 6 seems to be contrary to the conventional understanding, why higher intra-product international specialization in labor-intensive industries will reduce carbon emissions? and the higher intra-product international specialization in capital-intensive industries will increase carbon emissions? More convincing explanations (or literature support) should be given. Whether the different effects brought by the difference of scale effect, composition effect and technology effect under different capital intensity are reflected in the regression coefficient, if so, a complete coefficient display should be given.

5. Conclusion. Conclusion 1) is not supported by empirical results. In basic regressions, scale effect and technology effect are only included in regression as control variables, and it cannot be concluded that technology spillover and income growth are caused by international specialization within products.

good

Reviewer 3 Report

The article titled as “Carbon effects from intra-product international specialization: evidence from China’s manufacture industries” is very interesting and informative. However, the author (s) must incorporate the following issues to improve the article.

1.       In the abstract please mention explicitly one or couple of policy implications relevant to the study.

2.       On page 3, line 87-96, the author claimed some distinct features or contributions of this study from previous ones. The author must mention references or quote the studies from which this study is significantly different in all three contributions claimed.

3.       Please provide a table of recent literature relevant to the topic in chronological order.

4.       The theoretical framework is nicely built.

5.       Discussion of main results with contextualization i.e. consistency or contradiction with prior studies is very short. The author (s) needs to expand it.

6.       The policy implications should be drawn from obtained results and should be precisely linked with your study findings.

i have seen a few grammar mistakes. The authors should  proofread the article to avoid grammar and syntax errors. 

Reviewer 4 Report

Although this paper looks interesting, the following corrections are essential:

The title of the article does not reflect the content. The abstract should include the novelty of the article, conclusions and policy recommendations. The introduction should be reworked. The purpose and contributions of the study are not clear. No references were used in the introduction. Did the authors write the entire article from their own knowledge? In the introduction, the authors neglected the SDG statements. Authors can align this article with SDG goals by making use of the following articles. https://doi.org/10.1016/j.jenvman.2023.117317 https://doi.org/10.1016/j.renene.2023.01.080 https://doi.org/10.1002/sd.2383 https://doi.org/10.1016/j.gr.2022.05.002. There are many mathematical notations in the article and this hinders the fluency of the article. Show all detailed formulas in the attachment. What is the source of Figure 1? Table 3 is the main output of the article. But it is not explained in detail. Policy recommendations should address the findings of the study and the SDG. State the limits of the study.

The article should be examined in detail from the grammatical point of view.

Round 2

Reviewer 1 Report

I believe that the paper has gained in-depth interest. After revision, this paper has been highly improved. I suggest accepting this paper in its present form.

Reviewer 3 Report

The authors really worked hard to incorporate all comments. I am satisfied and the article is publishable now. 

Reviewer 4 Report

The efforts of the authors are sufficient for publication.

The efforts of the authors are sufficient for publication.